# A global survey of diurnal offshore propagation of rainfall

**Junying Fang** [1,2,3] **& Yu Du** [1,2,3] ✉

Diurnal rainfall offshore propagation (OP) shapes the timing and intensity of coastal rainfall and thus impacts both nature and society. Previous OP studies have rarely compared various coasts, and a consensus regarding physical mechanisms has not been reached on a global scale. Here, we provide the global climatology of observed OP, which propagates across ~78% of all coasts and accounts for ~59% of the coastal precipitation. Generally, OP is facilitated by low latitudes, high moisture conditions and offshore background winds. OP at low latitudes in a high-moisture environment is mainly caused by inertia–gravity waves due to the land–sea thermal contrast, whereas OP at higher latitudes is significantly influenced by background winds under trapped land–sea breeze circulation conditions. Slower near-shore OP might be modulated by density currents. Our results provide a guide for global OP hotspots and suggest relative contributions of mechanisms from a statistical perspective.

Diurnal variation constitutes an important component of weather and climate, with solar radiation as a primary driver[1–3]. Atmospheric temperature, wind, pressure, precipitation, and other meteorological variables are characterised by evident diurnal cycles, which are manifested as prominent geographic and seasonal distinctions[4,5]. The diurnal cycle is not only an important contributor to the evolution of meteorological variables on short time scales but is also closely linked to climate variability on long time scales[6–8]. The diurnal cycle of rainfall, as the fundamental mode of precipitation variability, provides opportunities to evaluate our understanding and modelling of the governing physical processes and multi-scale interactions[9]. Numerous previous studies have documented that rainfall generally peaks in the afternoon on land and in the nighttime over the ocean[10]. Thermal convection in the afternoon attributed to solar radiation is generally considered a main cause of the observed afternoon peak of rainfall on land, while cloud top nighttime radiation cooling increases thermal instability and thus promotes nocturnal rainfall over the ocean[2].

Interestingly, the most prominent diurnal variation in rainfall is manifested as phase propagation and often occurs on the leeward sides of mountains or plateaus (e.g., Rocky Mountains and Tibetan Plateau) or off various coasts (e.g., northwest coast of South America,

northeast coast of New Guinea, east coast of India, and South China). The diverse characteristics and complex mechanisms of diurnal rainfall propagation have attracted widespread attention. A number of studies have focused on diurnal rainfall propagation downstream of the Rocky Mountains and Tibetan Plateau and have attributed this phenomenon to convective systems originating from upstream mountains associated with the mountain-plains solenoid under the influence of background winds[11–15]. Some studies have also highlighted a vital role of density currents (cold pools) in the propagation and maintenance of rainfall[11–13,16]. Compared to the downstream region of mountains, relatively few studies have investigated diurnal rainfall offshore propagation (OP) due to observational limitations. However, in coastal areas where 37% of the global population lives[17], precipitation, including its diurnal variation, can strongly influence nature and weather-critical industries such as energy, logistics, fishery, and tourism and is regarded as an important subject.

With the development of remote sensing technology (e.g., satellites and radar) in recent years, OP has received increasing attention, but disagreements in the literature exist regarding the dominant processes and mechanisms (e.g., gravity waves and density currents) in different regions and even within the same regions. Yang and Slingo[18]

[1]School of Atmospheric Sciences, Sun Yat-sen University, and Southern Marine Science and Engineering Guangdong Laboratory (Zhuhai), Zhuhai 519082, China. [2]Guangdong Province Key Laboratory for Climate Change and Natural Disaster Studies, Sun Yat-sen University, Zhuhai 519082, China. [3]Key Laboratory of Tropical Atmosphere-Ocean System (Sun Yat-sen University), Ministry of Education, Zhuhai 519082, China. ✉e-mail: duyu7@mail.sysu.edu.cn

observed tropical OP through satellite observations in the Bay of Bengal, New Guinea, and Mexico, and they hypothesised a gravity wave mechanism triggered by diurnal variations in land and sea heat. This hypothesis has been confirmed via a series of subsequent observations, simulations and theoretical studies[15,19–29]. Mapes et al.[19] found that OP along the northwest coast of South America is mainly caused by gravity waves excited by thermal forcing of the coastal terrain with a speed of approximately 15 m s⁻¹. The associated gravity wave signal was observed via onboard instruments in the Joint Air–Sea Monsoon Interaction Experiment[30]. Gravity waves cause maximum convergence of sea surface winds 1–2 h earlier than the maximum precipitation[28]. Theoretically, inertia–gravity waves forced by the terrain/land–sea thermal contrast (hereafter referred to as IGWs) can be simulated well at low latitudes with a two-dimensional linear model of the land–sea breeze[15,26,27,31,32]. IGWs are essentially a linear component of land–sea breeze circulation at low latitudes, but no IGW signature is exhibited under trapped land–sea breeze circulation conditions at higher latitudes[26,31]. Additionally, gravity waves excited by convection on land comprise another important mechanism of offshore diurnal rainfall propagation, generating upward motion over the ocean causing mid-level cooling, thereby increasing instability and facilitating OP[21,25,29,33–38].

Precipitation systems under the influence of IGWs generally propagate outward at speeds ranging from 15 to 20 m s⁻¹ and travel across long distances, whereas certain coastal precipitation systems propagate at speeds from 3 to 7 m s⁻¹ within 100 km offshore associated with density currents (DCs), such as land breezes or cold pools[16,25,39–42]. The interaction between nocturnal land breezes and background winds can trigger convection near the land breeze front, and ensures that the nocturnal precipitation system propagates offshore with advancing density flow[33,39,43,44]. Land breeze fronts are essentially a nonlinear component of land–sea breeze circulation[45]. In addition, afternoon convection originating from land propagates towards the sea accompanied by cold pool outflow due to rain evaporation and can trigger new convection at the gust front with background winds, which also contributes to the nocturnal OP phenomenon[21,40].

Recent studies have illustrated similarities and differences in OP among multiple coasts of the Maritime Continent where differing relevant mechanisms occur[29]. However, most studies have focused on specific coastal regions in regard to the characteristics of OP and proposed specific views on mechanisms from a local perspective without a comparison to other coastal regions. To date, the characteristics and mechanisms of OP from a global perspective are not well understood. Since varying mechanisms may differ associated characteristics, the dominant mechanisms and key factors could be clarified by statistically characterising global OP phenomenon.

Under different dominant mechanisms, OP modulation may vary with environmental factors. For instance, background winds generate an advection effect on precipitation systems and thus dominate or modulate the propagation speed of these systems[21,46]. By contrast, background winds may regulate OP by altering the IGW pattern due to the Doppler effect[15,27,45,47]. The interaction between DCs and varying background winds favours convection triggering and maintenance and thus affects the propagation of precipitation systems. In addition, the properties of IGWs depend on the latitude[26], whereas DCs do not vary much with the latitude. A moist environment promotes favourable conditions for the diurnal cycle of rainfall, and this effect may vary under different mechanisms[25,41,48–54]. Therefore, it is necessary to statistically examine the responses of diurnal rainfall propagation to three key factors, including background winds, latitude, and moisture, which could help reveal the relative contributions of the various mechanisms and their global applicability.

In this work, we characterise the global climatological fingerprint of OP. Diurnal rainfall patterns off global coasts are examined over the period between 1998 and 2015 via satellite precipitation observations

(CMORPH) and reanalysis data (ERA5). Our results provide a preliminary guide for global OP hotspots in regard to risk evaluations or research interest in certain regions. Our results further suggest relative contributions of OP mechanisms from a statistical perspective around the globe.

## Results

### Global features of coastal diurnal rainfall patterns

The diurnal patterns of rainfall off global coasts can be grouped into four categories: OP, phase-locked (PL), shoreward propagation (SP), and weak or incoherent (WI) diurnal cycle (Fig. 1a–d). OP accounts for the highest occurrence percentage (~37%, Fig. 1e) and precipitation contribution (~59%, Fig. 1f) among the various diurnal patterns, indicating its major role in global coastal precipitation. During OP, the diurnal rainfall generally peaks at ~5:00 local solar time (LST) near coasts and peaks at ~9:00 LST far from coasts, manifesting offshore diurnal propagation features (Fig. 1a). The propagation distance varies greatly among samples with a largest distance of ~1400 km. The PL diurnal pattern, accounting for the second highest occurrence percentage (~28%, Fig. 1e) and precipitation contribution (~22%, Fig. 1f), exhibits diurnal phases concentrated during the early morning (~6:00 LST, Fig. 1b). The diurnal peak of rainfall during SP occurs from 4:00 LST (far from the shore) to 7:00 LST (near the shore), whose propagation direction is the opposite to that during OP (Fig. 1c). These three notable diurnal patterns all exhibit post-midnight or morning diurnal rainfall peaks offshore. The PL and SP diurnal patterns exhibit larger diurnal amplitudes in near-shore areas within a few tens of kilometres, while OP maintains a notable diurnal variation in both near- and far-shore areas.

Figure 1e shows the global occurrence distribution of diurnal rainfall patterns for each studied sample (see "Methods"). OP is widely distributed across the globe with ~78% of all coasts exhibiting OP samples. In general, OP occurs frequently in the tropics (<30°), where OP accounts for ~41% of all samples, especially in humid environments. In contrast, OP accounts for only ~26% of all samples at middle latitudes (30°–60°). At middle latitudes, OP tends to occur on the east coasts (~93%), while SP preferentially occurs on the west coasts (~82%). The majority of coasts exhibits a remarkable seasonal variation in diurnal patterns, while four coasts of western Colombia and the Maritime Continent exhibit OP year round. In seasonal statistics (Supplementary Fig. 1), the OP and SP ratios exhibit a remarkable inversely seasonal variation with a maximum of OP (SP) during the summer (winter). In the summer, OP accounts for the highest occurrence proportion (~51%) as well as the largest precipitation contribution (~73%) among various diurnal patterns (Fig. 1e, f). However, in the winter, OP only accounts for ~20% occurrence proportion and ~35% precipitation contribution (Fig. 1e, f).

The precipitation amount and diurnal amplitude vary among the above three diurnal rainfall patterns (Fig. 2). The mean hourly precipitation (see "Methods") during OP (0.18 mm) is nearly twice that during the SP or PL diurnal patterns, and the OP proportion significantly increases with the mean hourly precipitation, suggesting a key role of OP in rainfall, especially in extreme rainfall (Fig. 2b). Eight OP samples, shown in the top-right corner of Fig. 2a, all originate from the coastal water of western Colombia, which is one of the rainiest areas globally where nocturnal offshore propagating mesoscale convective systems frequently occur[19,55,56]. High positive correlations are found between the mean hourly precipitation and the mean diurnal amplitude of the OP (0.84), PL (0.83) and SP (0.89) patterns, where the slope of the least square fit line for the OP pattern (0.58) is much higher than that of the least square fit lines for the PL (0.32) and SP patterns (0.20). The mean diurnal amplitude during OP (0.08 mm) is also much higher than that during the SP or PL patterns (Fig. 2c). The above results indicate that rainfall during OP tends to be concentrated during several hours of the day. Therefore, OP exhibits a significant diurnal variation contributing remarkably to local precipitation.

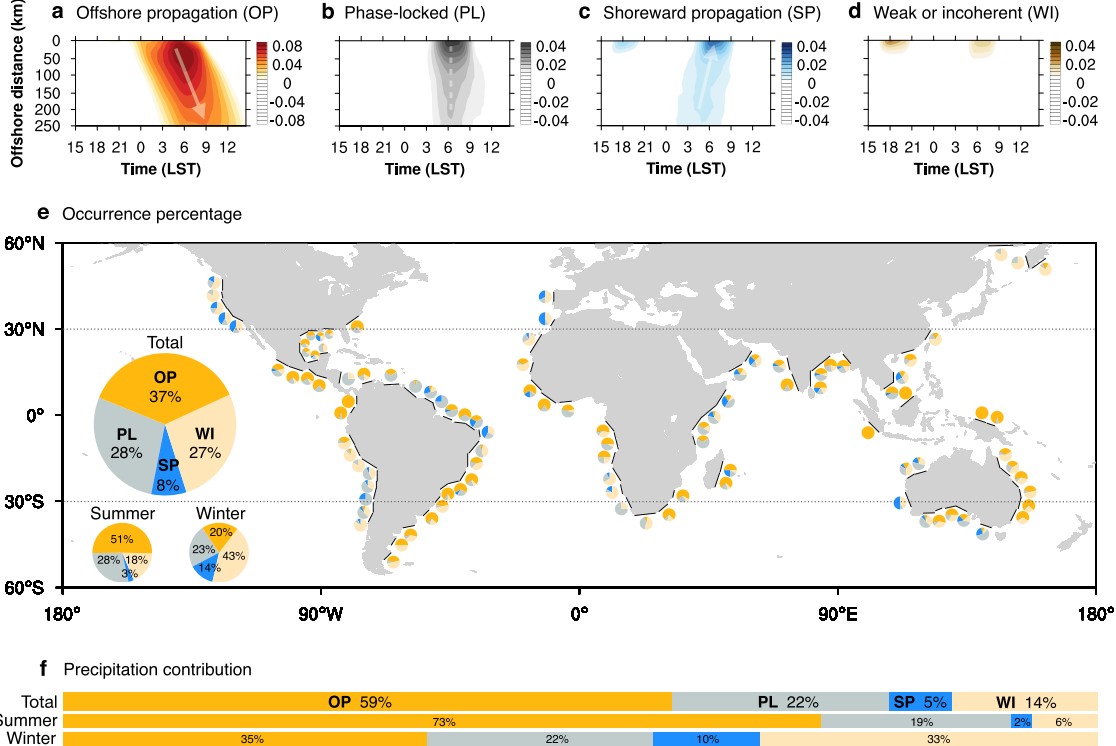

**Fig. 1 | Classifications and distributions of the coastal diurnal rainfall patterns.** **a**–**d** Composite distance–time Hovmöller diagrams of the hourly precipitation deviation (shaded, mm) for each type. **e** The global occurrence distribution and total percentage of each diurnal rainfall pattern off selected coastlines. The small pie charts indicate the occurrence percentage of each type for corresponding coast, and the big pie charts indicate the total occurrence percentage of each type with seasonal differences. Summer (winter) is defined as JJA (DJF) in the Northern Hemisphere and DJF (JJA) in the Southern Hemisphere. One hundred and two selected coastlines are indicated by black lines. **f** The precipitation contribution for each type with seasonal differences. The Hovmöller diagrams and map were generated using the NCAR Command Language (Version 6.6.2) [Software]. (2019). Boulder, Colorado: UCAR/NCAR/CISL/TDD. https://doi.org/10.5065/D6WD3XH5.

## Influences of environmental factors

Previous studies have suggested that background winds, latitude and moisture should be considered important factors regulating OP near specific coasts, as mentioned in the Introduction section[15,21,25–27,41,45,47–54]. This section further elucidates the statistical relationships between diurnal rainfall propagation off global coasts and corresponding controlling factors.

Figure 3a shows that the proportion of each diurnal type varies with the coast–normal component of the background wind averaged within 850–300 hPa (cloud layer, see "Methods"). Generally, the proportion of downwind propagation (~35%) is much higher than that of upwind propagation (~9%), especially under strong background winds. These results suggest that the advection effect of background winds plays a significant role in diurnal propagation. Previous studies[21,52,57] have also found that diurnal precipitation propagation significantly depends on advection processes of near-shore clouds and convection. However, the number of OP samples is more than 4 times that of the SP samples (Fig. 1), with samples of offshore background winds roughly equivalent to those of onshore background winds, which indicates that background winds might not constitute the only dominant factor of diurnal propagation and that there might occur additional mechanisms contributing to the direction of propagation. Under strong (>4 m s⁻¹) downwind conditions, OP accounts for ~57%, while SP accounts for only ~21% of all samples. Under weak (<4 m s⁻¹) downwind conditions, the proportion of OP (~52%) even reaches ~4 times that of SP (~12%). Under weak upwind conditions, SP accounts for only ~2%, but the proportion of OP can still reach ~27% of all samples, which is even much higher than that of SP under weak downwind conditions (~12%). Besides, a few OP samples even occur under strong upwind conditions, but none of SP samples occur under those conditions.

Since the majority of the OP samples (~73%) is concentrated under weak background wind conditions, other mechanisms in addition to the advection effect might greatly impact OP.

The effect of the latitude on diurnal propagation is further examined in a statistical manner. As shown in Fig. 3b, the proportion of OP reaches ~40% at low latitudes (<30°, including both the Northern and Southern Hemispheres) and is much higher (>5 times) than that of SP. In contrast, the ratio of OP is only ~14% at higher latitudes (>35°), which reaches the same order of magnitude as that of SP (~8%). The number distribution of OP indicates that ~82% of the OP samples are concentrated in low-latitude areas. Previous studies regarding notable OP near specific coasts were also generally confined within the tropics (30°N–30°S), such as western Colombia[19], the Maritime Continent[21], the Bay of Bengal[15], and the South China Sea[27]. The dependence of diurnal patterns on the latitude suggests an important role of IGWs[26], which will be examined in the next section.

In addition to background winds and the latitude, a sufficiently humid environment is conducive to an enhanced diurnal variability of precipitation[25,41,48–54]. Figure 3c shows the global statistical relationship between moisture and diurnal rainfall propagation. The OP and SP ratios are comparable under insufficient moisture conditions. With increasing moisture condition, OP gradually increases and becomes much more abundant than SP. Under sufficient moisture (specific humidity > 11 g kg⁻¹), the ratio of OP reaches up to approximately 74%, and almost no SP occurs (~1%). Despite similar mean background wind and latitude in the summer and winter samples, the mean 850-hPa specific humidity is ~45% higher in the summer (8.5 g kg⁻¹) than winter (5.9 g kg⁻¹), which is a possible cause of the significant seasonal variation in OP occurrence proportion (51% vs. 20%, Supplementary Table 1). Du and Rotunno[27] found that vertical vapour advection

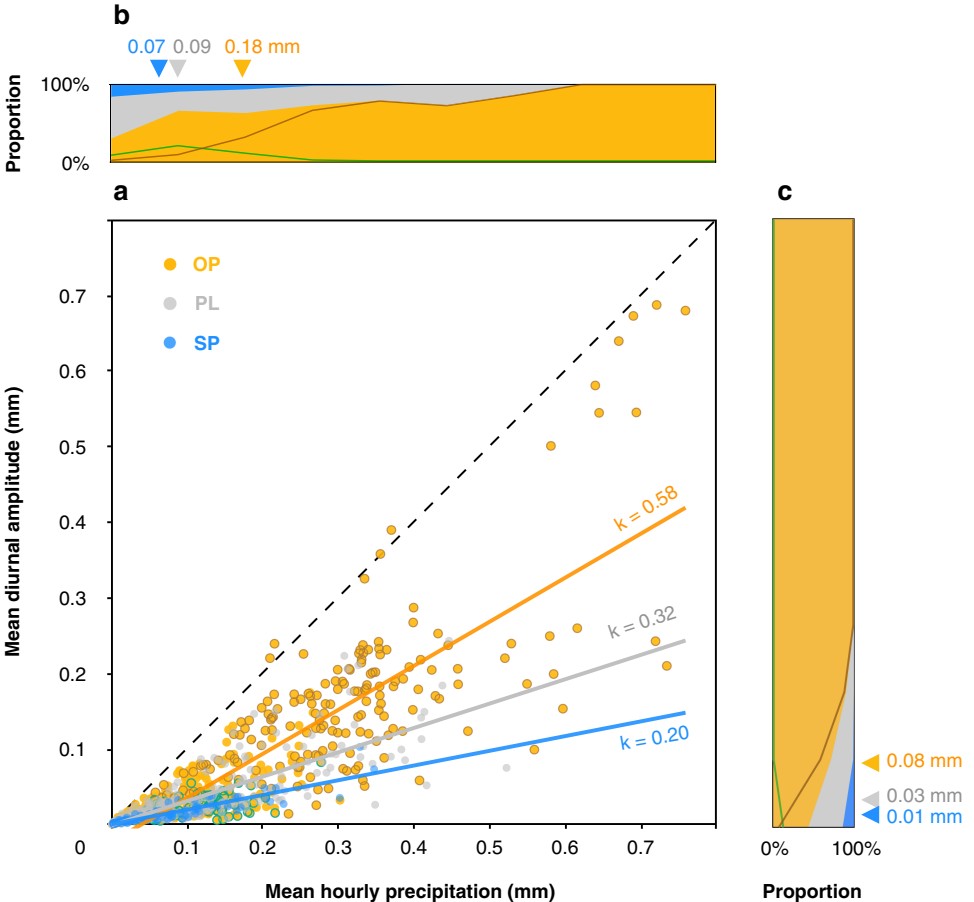

**Fig. 2 | Precipitation and the diurnal amplitude per diurnal pattern. a** Scatter diagram of the mean diurnal amplitude of rainfall versus mean hourly precipitation. OP, PL, and SP denote offshore propagation, phase-locked and shoreward propagation diurnal rainfall patterns, respectively. The orange dots with brown contours indicate OP samples with sufficient moisture ($q > 11 \, g \, kg^{-1}$). The orange dots with green contours indicate OP samples at higher latitudes (>30°). The weak or incoherent (WI) diurnal cycle is not shown. The mean hourly precipitation of each sample was calculated within the rectangle enclosed by the selected coastline and its offshore direction (perpendicular to the coastline) up to 250 km. The diurnal amplitude was derived from the amplitude of the first harmonic (diurnal cycle) in Fourier analysis. The black dashed line indicates that the mean hourly precipitation equals the diurnal amplitude. The least square fit lines between the hourly precipitation and diurnal amplitude of the OP (orange line, $r = 0.84$, $P < 0.001$), PL (grey line, $r = 0.83$, $P < 0.001$), and SP diurnal patterns (blue line, $r = 0.89$, $P < 0.001$) are indicated with the slopes (k). During the computation of the confidence levels, the reduction of effective sample size due to the autocorrelations has been considered (see "Methods section"). **b** Graph of the diurnal pattern proportion with mean hourly precipitation (mm). **c** Graph of the diurnal pattern proportion with mean diurnal amplitude of rainfall (mm). The triangles indicate mean hourly precipitation or mean diurnal amplitude for each diurnal pattern. The brown lines in **b, c** indicate OP samples with sufficient moisture ($q > 11 \, g \, kg^{-1}$). The green lines in **b, c** indicate OP samples at higher latitudes (>30°).

provides a physical connection between IGWs and OP. Therefore, sufficient moisture is required so that IGWs can influence diurnal rainfall propagation. These results indicate that in a sufficiently moist environment, IGWs might constitute the dominant factor of OP rather than the advection effect.

## Mechanisms of pronounced offshore propagation

Given that high moisture conditions and low latitudes facilitate OP, IGWs could be a possible dominant mechanism according to theorical IGW research[26,27]. To examine this hypothesis globally, we further focused on OP samples in a sufficiently moist environment ($q > 11 \, g \, kg^{-1}$) characterised by pronounced OP (hereafter referred to as POP). POP accounts for ~39% of the OP samples and consists of most OP samples featuring high precipitation and/or a large diurnal amplitude (the orange dots with brown contours in Fig. 2a and the brown lines in Fig. 2b, c). POP also includes almost all of the OP samples considered in previous studies, such as South China in June[26,27], western Colombia from May to December[15,19,55,56], New Guinea year round[25,29,34], the Bay of Bengal from June to September[15,18,28], Borneo in December[39] and Sumatra year round[21,29,35,41]. As shown in Supplementary Table 1, the propagation distance of the summer POP (635 km) is much longer than

the winter POP (393 km). Additionally, all POP samples occur at low latitudes (<30°), and ~94% of the POP samples occur under weak background wind conditions (<4 m s⁻¹).

Rotunno[31] proposed that land–sea breezes theoretically occur in the form of IGWs at latitudes lower than 30°, while they occur in the form of trapped circulations confined to the vicinity of the coastline at latitudes higher than 30°. Therefore, IGWs theoretically contribute to the OP phenomenon only in the tropics[26]. Following the discussion presented by Rotunno[31], the dispersion equation for hydrostatic IGWs is:

$$\frac{k}{m} = \frac{\sqrt{\omega^2 - f^2}}{N}, \qquad (1)$$

where $\omega$, $k$ and $m$ are the frequency and horizontal and vertical wavenumbers, respectively, in the wave packet. The horizontal phase speed is thus:

$$c = \frac{\omega}{k} = \frac{\omega}{m} \frac{N}{\sqrt{\omega^2 - f^2}}, \qquad (2)$$

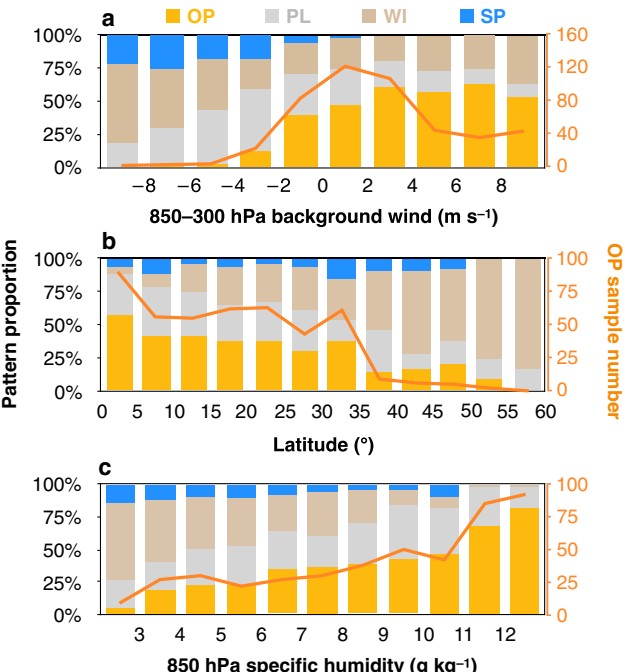

**Fig. 3 | Responses of the diurnal pattern proportion to environmental factors.** The environmental factors include **a** 850–300 hPa background wind, **b** latitude and **c** 850 hPa specific humidity. OP, PL, WI, and SP denote offshore propagation, phase-locked, weak or incoherent and shoreward propagation diurnal patterns, respectively. The corresponding sample number of OP is indicated by orange lines. Positive (negative) values of the averaged 850–300 hPa background wind indicate the offshore (onshore) component of the background wind. The background wind and specific humidity of each sample were calculated within the rectangle enclosed by the selected coastline and its offshore direction (perpendicular to the coastline) up to 250 km.

which indicates that the horizontal component of the phase velocity of the waves theoretically increases with the latitude within 0°–30° under the Coriolis effect (indicated by the brown triangles in Fig. 4a, which were derived using the simple 2D linear land–sea-breeze model with friction following Du and Rotunno[26], see "Methods").

In our statistics, the rainfall propagation speed in the POP samples (the red and yellow dots in Fig. 4a) indeed exhibits an increasing trend with the latitude ($r = 0.27$, $P < 0.001$), suggesting a certain association with IGWs. However, a quantitative discrepancy remains between the rainfall propagation speed in the POP samples and theoretical IGW solutions. Previous studies have suggested that IGW-dominated OP is usually initiated near the coastline in the early morning (after midnight) and then propagates offshore, which has been observed in western Colombia, western Mexico, eastern India, South China, etc[15,18,19,27,28]. Considering that the onset phase of the POP samples herein ranges from 18 to 09 LST, we further categorise POP into POP1 (71%) and POP2 (29%), where the onset phase starts after and before midnight, respectively. The propagation speed of POP1 ranges from 6 to 23 m s$^{-1}$ (the red dots in Fig. 4a), closer to that of the IGW-dominated OP phenomenon in the analytical solutions (the brown dots in Fig. 4a) and observations[15,18,19,27,28]. The increasing trend of the propagation speed with the latitude for POP1 (red line in Fig. 4a) is similar to that of the IGW analytical solutions (the brown line in Fig. 4a), which suggests that POP1 is greatly affected by IGWs. In contrast, POP2 exhibits a much lower propagation speed (4–15 m s$^{-1}$) with an increasing trend with the latitude (the yellow line in Fig. 4a), suggesting that POP2 might be affected by other mechanisms in addition to IGWs.

The composite Hovmöller diagrams of the hourly precipitation deviation between POP1 and POP2 are further compared (Fig. 4b–c).

POP1 exhibits a coherent diurnal propagation pattern starting from ~6:00 LST along the coast with a high speed (~13 m s$^{-1}$) from the near shore to the far shore. In contrast, POP2 exhibits slow propagation near the shore (~4 m s$^{-1}$) and fast propagation far from the shore (~13 m s$^{-1}$) with an earlier onset time (~21:00 LST) along the coast. These results further suggest that mechanisms other than IGWs mainly affect nearshore diurnal propagation in regard to POP2 (Fig. 4c). These features agree well with several previous studies focused on OP associated with DCs, such as land/mountain breeze fronts or convective cold pool outflows[21,25,39]. OP associated with DCs mostly propagates offshore before midnight and only extends to the near-shore area (~100 km) at a low speed (3–7 m s$^{-1}$), following previous studies worldwide through observations and theory[16,39–42]. Since the IGW-dominated OP phenomenon significantly differs from the DC-dominated OP phenomenon in terms of propagation distance and speed features, the entire propagation signal of POP2 comprises two steps, probably involving superposition of the DC mechanism near the coast and the IGW mechanism farther offshore[25,29].

Specifically, nearly half of the total POP samples exhibit a two-step diurnal propagation pattern. All POP2 samples are characterised by this two-step diurnal propagation pattern, with a speed transition at 90 km offshore on average, while only 29% of the total POP1 samples exhibit two-step propagation (two-step POP1). As shown in Fig. 4d, the far-shore (90–250 km) propagation speed of the POP2 samples (red circles) increases with the latitude, with their least square fit line (red line) significantly similar to that of the IGW solutions (brown line). In contrast, the near-shore (0–90 km) propagation speed of the POP2 samples (blue circles) exhibits a much lower propagation speed (~3–7 m s$^{-1}$) than that of the IGW solutions (~9–15 m s$^{-1}$, grey triangles). These results suggest that IGWs contribute more to POP2 in far-shore areas than in near-shore areas. In addition, the near-shore propagation speed of 3–7 m s$^{-1}$ for POP2 matches the DC characteristic speed (Fig. 4d). Such features are also found in the two-step POP1 (Supplementary Figs. 2 and 3). Hence, the two-step POPs (including all POP2 and two-step POP1 samples) is possibly caused by a combination of slow near-shore propagation dominated by DCs and fast far-shore propagation dominated by IGWs. IGWs emanate from the coast given an oscillatory heat source and produce a rainfall peak ~100 km offshore. Since DCs cannot propagate across great distances and weaken far from the shore, IGWs thus become dominant several hundred kilometres offshore in the two-step POPs process with a high propagation speed.

Additionally, the relationship between POPs and DCs is further explored through examining sensitivities of the advection effect of background winds (Supplementary Fig. 4). In the one-step POP1, the proportions of offshore and onshore background winds are comparable, indicating a limited role of the background wind advection effect. In contrast, the offshore background winds more often occur in the two-step POPs (83% in POP2 and 69% in two-step POP1). Such dependence on background wind direction is consistent with the advection properties of DCs. The near-shore propagation in the two-step POP1 starts latter than that in POP2 (~3:00 vs. ~21:00). The results above might suggest that both two-step POP1 and POP2 in the near-shore areas are related to DCs, specifically land/mountain breeze fronts[25,39,44,58] and cold pool outflows generated by afternoon land convection[40–42], respectively.

While the far-shore speed of POP2 matches that of IGWs well, the far-shore peak of POP2 occurs ~4 h earlier than that of POP1 (Fig. 4b–c). One possible explanation is the impact of gravity waves excited by convection[21,25,29,33–38]. Coastal convection of POP2 is probably stronger than that of POP1 in the evening, facilitating convectively generated gravity waves. These waves increase the atmospheric instability and facilitate OP before midnight and are superimposed with early morning IGWs on the climatology, thus resulting in POP2 exhibiting earlier far-shore propagation than that exhibited by POP1.

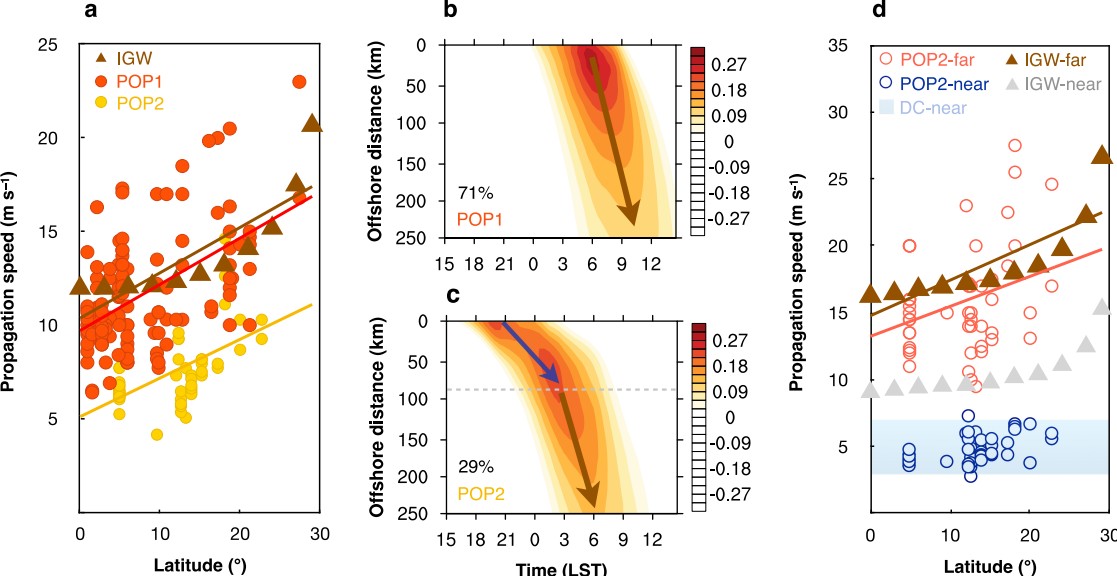

**Fig. 4 | Mechanisms of pronounced offshore propagation (POP). a** The propagation speed of the post-midnight POP samples (POP1, red dots), pre-midnight POP samples (POP2, yellow dots), and analytical solutions of inertia–gravity wave forced by the terrain/land–sea thermal contrast (IGW, brown triangles) with the latitude. The least square fit lines between the propagation speed and latitude for POP1 ($r = 0.53$, $P < 0.001$), POP2 ($r = 0.56$, $P < 0.001$) and IGWs ($r = 0.85$, $P < 0.005$) are indicated by yellow, red and brown lines, respectively. **b–c** Composite distance–time Hovmöller diagrams of the hourly precipitation deviation (shaded, mm) for POP1 and POP2. The brown and blue arrows indicate the diurnal rainfall propagation driven by IGWs and density currents (DCs), respectively. The propagation speed transition for POP2 is indicated by the grey dashed line. **d** The far-shore propagation speed of the POP2 samples (red circles) and IGW analytical solutions (brown triangles), and near-shore propagation speed of the POP2 samples (blue circles), IGW analytical solutions (grey triangles) and DC characteristic range (sky blue range) with the latitude. The least square fit lines between the far-shore propagation speed and latitude for POP2 ($r = 0.30$, $P < 0.05$) and IGWs ($r = 0.83$, $P < 0.005$) are indicated by red and brown lines, respectively. The POP samples are identified as OP samples with sufficient moisture ($q > 11$ g kg$^{-1}$). The Hovmöller diagrams were generated using the NCAR Command Language (Version 6.6.2) [Software]. (2019). Boulder, Colorado: UCAR/NCAR/CISL/TDD. https://doi.org/10.5065/D6WD3XH5.

## Discussion

The results above indicate that the proportions of the different diurnal propagation types vary with background winds, latitude, and moisture. Table 1 summarises their combined influences to elucidate their interactions.

In the samples (totally accounting for 20% of all samples) characterised by low latitudes (<30°) and sufficient moisture conditions ($q > 11$ g kg$^{-1}$), diurnal rainfall variation is evident, and the WI type barely occurs. OP dominates under offshore background winds (88%) or calm background winds (<1 m s$^{-1}$; 90%). OP remains in 37% of all samples, and SP barely occurs (3%) even under onshore background winds. In these situations, OP is characterised by remarkable precipitation and a large diurnal amplitude (the brown lines in Fig. 2b, c), and is mainly attributed to IGWs starting after midnight at a high speed (12–21 m s$^{-1}$) and propagating more than 100 km away from the coast, while background winds only play a modulating role (Fig. 5a). In nearly half of the OP samples, DCs might greatly facilitate earlier initiation of the near-shore OP process mostly before midnight, travelling across less than 100 km offshore at a low speed (3–7 m s$^{-1}$), after which IGWs become dominant, thereby exhibiting a two-step diurnal propagation pattern (Fig. 4c).

In contrast, at higher latitudes (totally accounting for 26% of all samples), the diurnal rainfall variation is relatively weak and the percentage of the WI type ranges from 36 to 58% (Table 1). The diurnal propagation direction greatly depends on the background wind direction, and diurnal propagation becomes nonsignificant (11% of the OP samples and 11% of the SP samples) under calm background winds. In these situations, OP is characterised by weak precipitation and diurnal cycle (the green lines in Fig. 2b, c), and is mainly attributed to the advection effect of background winds (Fig. 5b). The land–sea breeze circulation becomes trapped without IGWs, while DCs play a modulating role.

In the samples at low latitudes but with insufficient moisture (totally accounting for 54% of all samples), the proportion of each diurnal pattern varies between those in the above two situations, suggesting that IGWs and background winds likely jointly modulate OP, and their interaction could be complicated because background winds not only influence the advection of convection but also influence the pattern of IGWs due to Doppler shifting[27].

In addition to the discussion above, it should be noted that these three factors are not completely independent. For instance, background winds tend to strengthen with the latitude, while moisture tends to decrease with the latitude. Therefore, sufficient moisture generally coincides with weak background winds in the tropics, which favours the IGW-dominated OP phenomenon. Additionally, winds and moisture can vary under climate change and may thus cause a shift in diurnal patterns, which will be explored in the future. In particular, offshore storms may occur more frequently during the working hours of various industries on certain coasts and cause severe impacts.

Since previous studies[15,19] highlighted the role of coastal terrain in OP, we also examined the terrain effects in statistics. Although no clear relationship between coastal terrain height and OP occurrence proportion is found, coastal terrain can modulate the characteristics of OP. In general, large-range coastal topography (within 500 km onshore) largely affects the strength of the OP, probably due to the enhanced IGWs forced by the superposition of terrain and land–sea thermal contrast under the influence of large-range coastal topography. In contrast, small-range coastal topography (within 100 km onshore) induces earlier onset phase and slower near-shore propagation speed of OP, which is possibly associated with the enhanced DCs near shore under the influence of small-range coastal terrain. Due to the article length limit of this study, we will further investigate the interesting issue in detail in a separate work in the future.

**Table 1 | Combined influences of the environmental factors on the diurnal pattern proportion**

| Latitude | Moisture | Wind | OP (%) | PL (%) | SP (%) | WI (%) | Amount | Mechanisms |
|---|---|---|---|---|---|---|---|---|
| <30° | sufficient | offshore | 88 | 8 | 1 | 3 | 99 | Inertia–gravity wave-dominated |
|  |  | calm | 90 | 10 | 0 | 0 | 72 |  |
|  |  | onshore | 37 | 60 | 3 | 0 | 68 |  |
|  | insufficient | offshore | 51 | 21 | 1 | 27 | 258 | Both modulation effects |
|  |  | calm | 23 | 43 | 7 | 27 | 149 |  |
|  |  | onshore | 10 | 38 | 17 | 35 | 254 |  |
| >30° | insufficient | offshore | 51 | 15 | 0 | 35 | 150 | Advection effect-dominated |
|  |  | calm | 11 | 11 | 11 | 68 | 19 |  |
|  |  | onshore | 3 | 24 | 24 | 49 | 155 |  |

Samples at latitudes higher than 30° and with sufficient moisture are not shown due to their absence. Sufficient (insufficient) moisture is defined as 850-hPa specific humidity higher (lower) than 11 g kg$^{-1}$. Offshore/onshore winds are defined as averaged 850–300 hPa offshore/onshore background wind components with a speed higher than 1 m s$^{-1}$, and the remainder includes calm winds. OP, PL, SP and WI denote offshore propagation, phase-locked, shoreward propagation and weak or incoherent diurnal patterns, respectively. The percentage numbers indicate the ratio of the corresponding pattern to the total patterns. Inertia–gravity waves indicate inertia–gravity waves forced by the terrain/land–sea thermal contrast.

**a** Diurnal rainfall offshore propagation at low latitudes

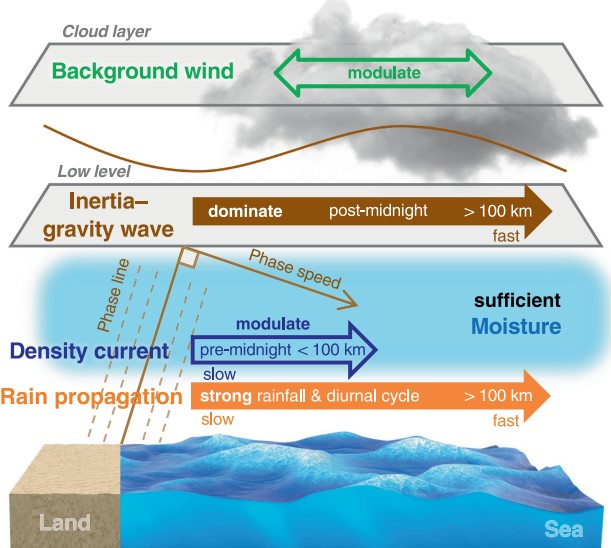

**b** Diurnal rainfall offshore propagation at middle latitudes

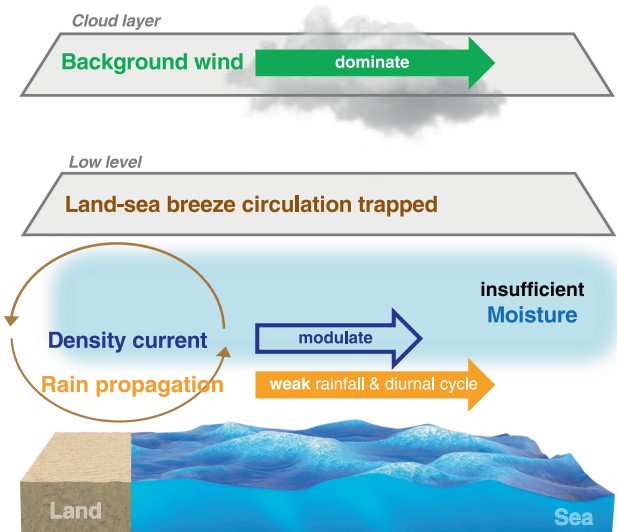

**Fig. 5 | Conceptual model of diurnal rainfall offshore propagation (OP). a** OP at low latitudes in a high-moisture environment is characterised by strong rainfall and diurnal cycle, and is mainly caused by inertia–gravity waves forced by the terrain/ land–sea thermal contrast. **b** OP at middle latitudes is characterised by weaker rainfall and diurnal cycle, and is significantly influenced by the direction of background winds under trapped land–sea breeze circulation conditions.

The analysis of the global OP features and mechanisms in the present study represents a key step in studying case-by-case OP samples and their impact. Our results (1) provide a preliminary guide for global OP hotspots for risk evaluations or research interest in certain regions, (2) highlight the importance of OP in regard to diurnal amplitude and precipitation contribution, and (3) reveal the relative contributions of different OP mechanisms from a statistical perspective globally. Weather and climate models that capture statistical features with exact physical mechanisms could provide decision-makers with accurate OP forecasts, especially to explore OP-related risks under a changing climate.

## Methods
### Criteria for the selection of coastlines
We adopted the following five criteria to select coastlines across the globe: (1) the selected coastlines should evenly cover the globe to ensure statistical significance. (2) The shape of the selected coastlines should be sufficiently straight with an irregularity less than 50 km, given that the land–sea breeze circulation is complicated near coasts with complex shapes[59,60]. (3) The length of the selected coastlines

should be greater than 250 km to obtain clearer and more reliable diurnal precipitation characteristics and other meteorological variables by averaging along the coastline[27]. (4) Coasts accompanied by opposing shores within 800 km offshore are not included to avoid the influence of these opposite coasts[29,43,61]. (5) Regarding islands or peninsulas, candidate coastlines should be selected where the width of the land region is larger than 250 km to obtain more prominent propagation characteristics of rain events[62]. Based on the above criteria, 102 coasts were identified globally, as shown in Fig. 1e.

### Data
We used a satellite-derived precipitation product, the bias-corrected version of the CPC MORPHing technique data (CMORPH)[63,64], from 1998 to 2015. CMORPH is available at a 30-min time resolution and a maximum horizontal resolution of 8 km globally (60°S–60°N). CMORPH can capture and quantify sub-daily variations in precipitation, allowing potential applications in examining diurnal cycles. Note that CMORPH has a negative bias at mid to high latitude land areas in the cool season due to unsatisfactory performance in detecting cold season precipitation and snowfall. Since OP is mainly distributed in

tropical seas as well as during warm season, the above effects on the results is limited.

The background winds and specific humidity offshore were obtained from ERA5 monthly data. ERA5 is a global atmospheric reanalysis product produced by the European Centre for Medium–Range Weather Forecasts (ECMWF) with a spatial resolution of 0.25°[65]. ERA5 achieves a good performance regarding wind and moisture fields, especially in the tropics. For a better option of the level of background winds, we conducted sensitivity tests on the effect of different levels of background wind (including 850 hPa, 700 hPa, 500 hPa, 300 hPa and 850–300 hPa) on the diurnal propagation of rainfall. The results show that the layer of 850–300 hPa is a good indicator for the wind advection effect, which is also used widely as the cloud-layer wind advcting storm cells in the previous studies[66–68]. The mean flow of the 850–300 hPa was calculated from winds averaged within 14 levels between 850 hPa and 300 hPa in ERA5 reanalysis data.

We mainly focused on the offshore coastal region within 250 km because this region, defined as typical coastal waters, is affected by coastal phenomena such as land–sea breeze circulation systems[69]. The studied distance (250 km offshore) contained the effect of both IGWs (within ~1000 km) and DCs (within ~100 km) offshore, which allowed us to compare their characteristics and contributions. The mean hourly precipitation, background winds and specific humidity of each sample were calculated within the rectangle enclosed by the selected coastline and its offshore direction (perpendicular to the coastline) up to 250 km.

### Global classification methodology of offshore diurnal rainfall

Considering that the diurnal rainfall pattern exhibits significant seasonal variations[2], each coast contains 12 studied samples on a monthly basis. A total of 1224 samples was thus identified involving 102 coasts within 12 months. To construct each sample, diurnal variation of precipitation at a given coast was averaged during given months of 1998–2015 (e.g., June 1998–2015). Each sample contains diurnal cycle information, and such process can filter the influence of synoptic systems to highlight the local diurnal cycle signal. Fourier analysis of rainfall was applied to identify the various types of diurnal propagation. The samples were classified as the WI type when the percent variance explained by the first harmonic (diurnal cycle) in Fourier analysis is lower than 50%. The remaining samples were further divided into three types. The samples were classified as PL samples when the diurnal phase (including the peak and zero phases) exhibits no significant shifting (<3 h) along 250 km offshore in distance–time Hovmöller diagrams of the hourly rainfall deviation. Otherwise, the samples were categorised as OP or SP samples depending on the propagation direction.

### Quantifying the characteristics of OP

The characteristics of OP were quantified with an objective method. The middle time of the zero phase at the coastline (0 km offshore) in Hovmöller diagrams was used as an estimate of the OP onset phase. The OP distance was estimated based on the offshore maximum distance where the coherent positive amplitude decreases to zero along the propagation direction or begins to change into the phase-locked pattern. Following previous studies[15,19,21,25,28,37,42,46], the OP speed was estimated via comprehensive combination of the diurnal peak and zero phases. When the propagation speeds using the diurnal peak and zero phases were generally consistent, the OP speed was obtained by averaging these two propagation speeds. Otherwise, the propagation speed with more significant phase shifting was selected as the OP speed.

### Linear land–sea-breeze model

The analytical solutions of theoretical horizontal component of the phase velocity of the IGWs within 0°–30° were derived using the simple 2D linear land–sea-breeze model with friction following Du and

Rotunno[26], which contains the 2D linear equations of motion under the Boussinesq and hydrostatic approximations with considering simplified frictional force. The heating forcing in the thermal equation is specified as:

$$Q = Q_0 \left( \frac{\pi}{2} - \tan^{-1} \frac{x}{x_0} \right) e^{-\frac{z}{z_0}} e^{-i\omega t}, \tag{3}$$

where the horizontal scale and the vertical scale of the land–sea contrast in heating are denoted by $x_0 = 50$ km and $z_0 = 1$ km, respectively. The dominant vertical wavelength in the analytical solutions of IGWs is closely related to the setting of $z_0$ in the model. The $\omega$ equals $2\pi$ day$^{-1}$, and $Q_0$ is the maximum heating rate. The coastline is at $x = 0$.

### Statistics

Correlations were calculated using Pearson's correlation coefficient defined as:

$$r = \frac{\sum_{i=1}^{n} (x_i - \bar{x})(y_i - \bar{y})}{\sqrt{\sum_{i=1}^{n} (x_i - \bar{x})^2} \sqrt{\sum_{i=1}^{n} (y_i - \bar{y})^2}}, \tag{4}$$

where $\bar{x}$ and $\bar{y}$ are the sample mean values of $x_i$ and $y_i$, respectively, from $i = 1$ to $i = n$ (n is the sample size). We determined the statistical significance level based on Student's t-test. Here, t is defined as:

$$t = r \sqrt{\frac{n-2}{1-r^2}}, \tag{5}$$

where $n$ can be replaced by the effective sample size[70] during the computation of the confidence levels to take into account the effect of autocorrelation in Fig. 2. The effective sample size ($N$) is estimated by:

$$N = n \frac{1 - r_1 r_2}{1 + r_1 r_2}, \tag{6}$$

where $r_1$ and $r_2$ are the lag-one autocorrelations of each series, respectively.

## Data availability

The CMORPH satellite precipitation data used in this study are available from the National Oceanic and Atmospheric Administration Climate Prediction Center through ftp.cpc.ncep.noaa.gov/precip/ CMORPH_V1.0/CRT. The ERA5 reanalysis data used in this study are available from the Copernicus Climate Data Store at https://cds. climate.copernicus.eu/. The data generated during the study are available from the corresponding author upon request.

## Code availability

The data processing scripts are available from the corresponding author upon request.

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

## Acknowledgements

This study was supported by the Guangdong Major Project of Basic and Applied Basic Research (2020B0301030004, Y.D.), the National Natural Science Foundation of China (Grant Nos. 42075006, 42122033, and 41875055, Y.D.), and Guangzhou Science and Technology Plan Projects (202002030346, Y.D.).

## Author contributions

Y.D. conceived the study. J.F. performed the analyses and wrote the paper. Y.D. and J.F. contributed to interpretation of the results and development of the paper.

## Competing interests

The authors declare no competing interests.
