## [Peer Review File · Nature Communications]

A global survey of diurnal offshore propagation of rainfallREVIEWER COMMENTS

Reviewer #1 (Remarks to the Author):

Review of "A Global Survey of Diurnal Offshore Propagation of Rainfall" by Fang and Du

Summary:

The authors investigated the characteristics of the diurnal rainfall offshore propagation (OP) and their associated mechanisms from a global perspective. To my knowledge, it is a first piece of work to provide a global climatology of OP while previous studies have been merely focused on the OP near specific coasts. The authors found that OP was greatly influenced by latitudes, moisture, and background winds, and the responsible mechanisms were different between lower and higher latitudes. The topic and results are interesting, and the manuscript is well organized, logical and easy to follow. I believe that the study makes a useful contribution to the body of literature on diurnal rainfall. Therefore, I recommend that the current manuscript be accepted for publication in Nature Communications after some minor revisions.

Specific comments:

- 1. The terrain near the coasts should be an important factor that influences OP through modulating land-sea breeze, as well as the gravity waves. The authors should consider this factor in their analysis. I understand that terrain is a complicated factor to address in this limited manuscript and it may better be investigated in a separate analysis. However, the authors should at least provide a reasonable discussion of the relevant subject.**
- 2. Is there any seasonal preference for OP? For example, how are the features revealed different between summer and winter?**
- 3. Line 165: The authors need to explain why 500 hPa has been chosen as the level for discussion of the background winds.**
- 4. Line 225: How to determine the value of m when calculating the horizontal phase speed c based on Eq. (2)?**
- 5. Lines 350-354: The errors of the CMORPH need to be indicated herein.**
- 6. Figure 2: There is no figure number for the subplots. No units can be found for the values with the triangles.**
- 7. Figure 3: There are no units for latitude.**
- 8. Figure 4: In the figure caption, 'D' should be in the lower case.**
- 9. Figure 5: It is hard to read the words on the yellow arrows.**

Reviewer #2 (Remarks to the Author):

Overview: This study used satellite and reanalysis data to investigate diurnal offshore precipitation propagation on a global scale. Previous studies have typically focused on offshore propagation over one or a very few coasts therefore lack agreement from place to place. This study attempted to analyze the offshore precipitation characteristics depending on latitude, moisture and background wind from a globally unified perspective. The authors provided solid quantitative analysis to the inertia-gravity waves (IGW) and density current (DC), which adds up to the attempts of the community to reach a consensus in terms of physical mechanisms. I really appreciate the part where this study divided pronounced offshore propagation (POP) into pre-midnight and post-midnight samples, which helps explain the long-lasting discrepancy among previous studies where the offshore propagation speeds near shore are different based on different observations. However, there are a few issues that the authors may need to give additional analysis to make the arguments more concrete. The authors were also unclear about some methodology details. Overall, this study is interesting, well-written and of importance to our community. It could be considered to publish on Nature Communications after the following issues are addressed.

1. L261-264: The authors found that 29% of the total POP1 samples have two-step propagation. Since POP1 is defined as post-midnight samples, I would assume that its slower nearshore propagation starts after midnight. Then the questions are: Do the two-step POP1 samples have a nearshore propagation speed range similar to POP2 (i.e., 3-7 m/s)? If they do, is it related to DCs (because the authors mentioned in L254-255 that DCs generally propagate offshore before midnight)? If they don't, what causes the two-step POP1 events?

2. Fig. 2 and L396: The authors calculated the least square fit between the hourly precipitation and the diurnal amplitude of the offshore propagation. One thing to note is that time series data may have temporal autocorrelation. Therefore, when calculating least square fit using time series data, the effective sample size may not equal to 'n' (Bretherton 1999), leading to different t and p values. This study used precipitation data, which typically have a very low temporal autocorrelation, so I would say that the results should not change much, especially when the p-values are so small. But a few words on this issue would be good just for prudence.

References:

Bretherton, C.S., Widmann, M., Dymnikov, V.P., Wallace, J.M. and Bladé, I., 1999. The effective number of spatial degrees of freedom of a time-varying field. *Journal of climate*, 12(7), 1990-2009.

3. Some unclear technical/methodology details:

1) L138-139: The authors only described how they identified offshore propagation with the hourly precipitation data in the method section. But how did they determine whether a month had offshore propagation or not using monthly sample?

2) L145-146: How did the author calculate the mean hourly precipitation for the four categories? Specifically, up to what distance offshore/onshore did the authors average the precipitation? Is the area averaged the same for the four categories?

3) Fig.3: Over how large the area did the authors define the background wind and specific humidity?

Minor comments:

1. L139-142: It is unclear what the percentages in the parentheses are. Are they percentages of the time or coasts?

2. Fig. 2: The orange dots with green contours are so overlapped with other dots. It would be good to make the dots of other colors more transparent (or use other better ways).

Response to Comments of Reviewer #1

Reviewer #1:

Review of “A Global Survey of Diurnal Offshore Propagation of Rainfall” by Fang and Du

Summary:

The authors investigated the characteristics of the diurnal rainfall offshore propagation (OP) and their associated mechanisms from a global perspective. To my knowledge, it is a first piece of work to provide a global climatology of OP while previous studies have been merely focused on the OP near specific coasts. The authors found that OP was greatly influenced by latitudes, moisture, and background winds, and the responsible mechanisms were different between lower and higher latitudes. The topic and results are interesting, and the manuscript is well organized, logical and easy to follow. I believe that the study makes a useful contribution to the body of literature on diurnal rainfall. Therefore, I recommend that the current manuscript be accepted for publication in Nature Communications after some minor revisions.

We appreciate the reviewer’s comments and suggestions to significantly improve our manuscript. Our point-by-point response is given in blue as follows.

Specific comments:

1. The terrain near the coasts should be an important factor that influences OP through modulating land-sea breeze, as well as the gravity waves. The authors should consider this factor in their analysis. I understand that terrain is a complicated factor to address in this limited manuscript and it may better be investigated in a separate analysis. However, the authors should at least provide a reasonable discussion of the relevant subject.

As suggested, we gave a brief discussion on the terrain effect on OP in the revised manuscript. Previous studies (e.g., Mapes et al. 2003; Li and Carbone 2015) typically highlighted the topographic influences within 500 km of the coastline regarding OP dominated by inertia–gravity waves due to the land–sea thermal contrast (hereafter referred to as IGWs). We statistically examined the role of both small-range coastal terrain (within 100 km onshore) and large-range coastal terrain (within 500 km onshore) in the OP characteristics (Fig. R1). In statistics, the height of small-range coastal topography ranges from 6 to 3006 m, with an average of 250 m near the global coasts, while the height of large-range coastal topography ranges from 50 to 3104 m, with an average of 618 m.

As shown in Fig. R2, there is no clear relationship between the OP occurrence proportion and coastal terrain height including small-range terrain and large-range terrain, suggesting that topography is not a major factor controlling OP occurrence.

Fig. R1: Schematic diagram of small-range and large-range coastal topography.

Fig. R2: Responses of the diurnal pattern proportion to small-range and large-range coastal terrain height. OP, PL, WI and SP denote offshore propagation, phase-locked, weak or incoherent and shoreward propagation diurnal patterns, respectively. The corresponding sample number of OP is indicated by orange lines. Small-range (large-range) coastal terrain height indicates the coastal terrain height averaged within 100 km (500 km) onshore.

Nevertheless, coastal topography has a significant influence on the characteristics of OP, and small-range and large-range coastal terrain exhibit their different effects. As shown in Table R1, large-range coastal topography largely affects the strength of the OP. The OP near higher large-range coastal topography (higher than median) has stronger mean precipitation (~21% stronger) and diurnal amplitude (~100% stronger) compared to the lower topography. In addition, the mean propagation distance of pronounced offshore propagation (POP that is OP samples with specific humidity greater than 11 g kg^{-1}) near higher terrain is longer by 62% than that near lower terrain. The differences are possibly caused by the enhanced IGWs forced by the enhanced terrain/land–sea thermal contrast under the influence of topography.

Table R1 Influences of the coastal terrain height on OP features.

	Large-range topography		Small-range topography	
	Higher	Lower	Higher	Lower
OP samples amount	246	206	211	241
OP percentage of samples	40%	34%	35%	39%
OP percentage of precipitation	72%	48%	64%	68%
OP mean hourly precipitation (mm)	0.20	0.16	0.19	0.17
OP mean diurnal amplitude (mm)	0.11	0.05	0.08	0.09
POP samples amount	110	67	87	90
POP mean distance (km)	626	459	583	545
POP mean speed (m s^{-1})	10.6	10.2	9.5	11.3
POP mean onset phase (LST)	2	2	0	4
POP2 percentage of POP samples	29%	30%	52%	8%

The large-range (small-range) topography is defined as coastal terrain heights averaged within 500 km (100 km) of the coastline. Higher (lower) topography denotes coastal terrain heights higher (lower) than median, which is 399 m (119 m) for large-range (small-range) coastal topography. POP is pronounced offshore propagation with sufficient moisture ($q > 11 \text{ g kg}^{-1}$), and POP2 is pre-midnight POP.

In contrast, small-range coastal topography mainly affects the onset phase and near-shore propagation speed of POP (Table R1). The proportion of POP2 in POP samples near higher small-range coastal topography (~52%) is ~7 times greater than that near lower topography (~8%). Accordingly, POP near higher terrain starts ~4 h earlier and propagates ~2 m s^{-1} slower. When the height threshold for distinguishing between higher and lower small-range topography changes from 119 m (median) to 350 m (horizontal grey dashed line in Fig. R3a), the role of small-range topography in the onset phase can be further highlighted (Fig. R3a). POP2 account for only 7% of POP near terrain lower than 350 m, but 100% of POP near higher terrain (Fig. R3a). Interestingly, the Hovmöller diagram of POP near higher (lower) small-range topography (Fig. R3b, c) is highly similar to that of POP2 (POP1) (Fig. 4b, c). Therefore, higher small-range coastal topography is probably responsible for the enhancement of convective cold pool outflows by intensifying land convection in the afternoon or nocturnal land breeze and thus favours POP2, consistent with the important role of density currents (DCs) for near-shore propagation.

Fig. R3: The role of small-range coastal topography in the onset phase and near-shore speed of POP. **a** Scatter plot of POP onset phase and small-range coastal terrain height. The horizontal grey dashed line denotes 350 m. The yellow and red dots represent POP2 and POP1 samples respectively. **b** Composite distance–time Hovmöller diagrams of the hourly precipitation deviation (shaded, mm) for POP samples with small-range coastal topography higher than 350 m. **c** Same as **b** but for small-range coastal topography lower than 350 m. POP is pronounced offshore propagation with sufficient moisture ($q > 11 \text{ g kg}^{-1}$), and POP1 (POP2) is post-midnight (pre-midnight) POP.

Considering the limited manuscript, we added a brief description on the terrain effect on OP in the discussion section (Lines 345–354) of current manuscript, and will study the interesting issue in detail in a separate work in the future (in preparation).

References:

- Mapes, B. E., Warner, T. T. & Xu, M. Diurnal patterns of rainfall in northwestern South America. Part III: Diurnal gravity waves and nocturnal convection offshore. *Mon. Weather Rev.* **131**, 830–844 (2003).
- Li, Y. & Carbone, R. E. Offshore propagation of coastal precipitation. *J. Atmos. Sci.* **72**, 4553–4568 (2015).

2. Is there any seasonal preference for OP? For example, how are the features revealed different between summer and winter?

Thanks for your suggestions. In a month-by-month view, the OP and shoreward propagation (SP) ratios exhibit a remarkable inversely seasonal variation in both hemispheres (Fig. R4 or Supplementary Fig. 1). The OP proportion reaches its maximum (minimum) in the warm (cool) season while the SP proportion does the opposite.

Fig. R4: Responses of the diurnal pattern proportion to seasonal variation in each hemisphere. OP, PL, WI and SP denote offshore propagation, phase-locked, weak or incoherent and shoreward propagation diurnal patterns, respectively. Fifty-one coasts were selected in each hemisphere.

Generally, in the summer (June–August in the Northern Hemisphere and December–February in the Southern Hemisphere) OP accounts for the highest occurrence proportion (~51%) as well as the largest precipitation contribution (~73%) among various diurnal patterns (Fig. R5a). In the winter (December–February in the Northern Hemisphere and June–August in the Southern Hemisphere), however, OP only accounts for ~20% occurrence proportion and ~35% precipitation contribution (Fig. R5b). Furthermore, the global distribution of OP occurrence also shows a remarkable seasonal variation. In the summer, OP is widely distributed with ~65% of all coasts across the globe exhibiting OP samples. But, in the winter only ~35% of global coasts present OP samples. Despite similar mean background wind and latitude in the summer and winter samples, the mean 850-hPa specific humidity is ~45% higher in the summer (8.5 g kg^{-1}) than winter (5.9 g kg^{-1}), which is a possible cause of the significant seasonal variation in the OP ratio (Table R2).

Fig. R5: Seasonal differences and global distributions of the coastal diurnal rainfall patterns. Summer (winter) is defined as JJA (DJF) in the Northern Hemisphere and DJF (JJA) in the Southern Hemisphere. The sample size for both seasons is 306. OP, PL, WI and SP denote offshore propagation, phase-locked, weak or incoherent and shoreward propagation diurnal patterns, respectively. The small pie charts indicate the occurrence percentage of each type for corresponding coast, and the big pie charts indicate the total occurrence percentage of each type. The horizontal histograms indicate the precipitation contribution for each type.

The characteristics of OP also differ in the summer and winter (Table R2). For instance, the summer OP has significantly stronger precipitation rate (0.21 vs. 0.15 mm h^{-1}) with larger diurnal amplitude (0.10 vs. 0.07 mm) compared to the winter OP. As for POP, the propagation distance in the summer (635 km) is much longer than in the winter (393 km), which might be related to the enhancement of IGWs due to stronger land–sea thermal contrast in the summer.

Table R2 Seasonal differences in OP features.

	Summer	Winter
Mean background wind (m s^{-1})	0	0
Mean latitude ($^{\circ}$)	21	21
Mean specific humidity (g kg^{-1})	8.5	5.9
OP samples amount	155	61
OP percentage of samples	51%	20%
OP percentage of precipitation	73%	35%
OP mean hourly precipitation (mm)	0.21	0.15
OP mean diurnal amplitude (mm)	0.10	0.07
POP samples amount	67	19
POP mean speed (m s^{-1})	11.2	10.1
POP mean onset phase (LST)	2	3
POP mean distance (km)	635	393

Summer (winter) is defined as JJA (DJF) in the Northern Hemisphere and DJF (JJA) in the Southern Hemisphere. Positive (negative) values of the background wind indicate the offshore (onshore) component of the averaged 850–300 hPa background wind. The specific humidity takes its value at 850 hPa. OP denotes diurnal rainfall offshore propagation. POP is pronounced offshore propagation with sufficient moisture ($q > 11 \text{ g kg}^{-1}$).

The related Fig. 1, Supplementary Fig. 1 and the text at Lines 143–148, 204–207, 223–224 were updated in the revised manuscript.

- Line 165: The authors need to explain why 500 hPa has been chosen as the level for discussion of the background winds.

Thanks for your suggestions. We adjusted the level of background winds from 500 hPa to 850–300 hPa and clarified it in the revised manuscript.

Generally, the mean flow within 850–300 hPa is usually considered to be the cloud-layer wind advecting storm cells (e.g., Newton and Fankhauser 1964). For coastal rainfall, previous studies also highlighted the role of offshore mid-troposphere wind in advecting rainfall offshore (e.g., Wei et al. 2020).

For a better option of the level of background winds, we conducted sensitivity tests on the effect of different levels of background wind (including 850 hPa, 700 hPa, 500 hPa, 300 hPa and 850–300 hPa) on the diurnal propagation of rainfall. Ninety percent of SP is under the downwind background at 500 hPa or 850–300 hPa, while the proportion is lower for the levels of 850 hPa, 700 hPa, and 300 hPa as background winds (Table R3). The results suggest the dominant role of background wind advection in SP. By contrast, the proportion of downwind condition at 500h Pa or 850–300 hPa in OP also reaches the highest (77%) compared to other levels but with a relatively low proportion (Table R3), suggesting the mechanisms of OP are more complex than those of SP.

Table R3 The effect of different levels of background wind on the diurnal propagation of rainfall.

	850 hPa	700 hPa	500 hPa	300 hPa	850–300 hPa
Sample size of downwind OP	256	335	348	319	346
Sample size of downwind SP	70	83	88	85	88
Downwind proportion in OP samples	57%	74%	77%	71%	77%
Downwind proportion in SP samples	71%	85%	90%	87%	90%

OP and SP denote offshore propagation and shoreward propagation diurnal patterns, respectively. The sample size of OP and SP are 452 and 98, respectively. The 850–300 hPa wind denotes wind averaged within 14 levels between 850 hPa and 300 hPa in ERA5 reanalysis data.

Therefore, both the level of 500hPa and the layer of 850–300 hPa are good indications of the wind advection effect for the diurnal propagation of rainfall. We found all related results in this study remains high similarity whether 500 hPa or 850–300 hPa is adopted as the level of background wind. Since the layer of 850–300 hPa is used more widely in the previous studies (e.g., Chappell 1986; Corfidi et al. 1996; VandenBerg et al. 2014), we changed from 500 hPa to the layer of 850–300 hPa as the level of background wind in the revised manuscript.

The related Fig. 3, Fig. 5, Table 1 and the text at Lines 170–172, 179–186, 225–226, 314–316, 319, 327, 390–397 were updated in the revised manuscript.

References:

- Newton, C. W., & Fankhauser, J. C. On the movements of convective storms, with emphasis on size discrimination in relation to water-budget requirements. *J. Appl. Meteorol. Climatol.* **3**, 651–668 (1964).
- Wei, Y., Pu, Z. & Zhang, C. Diurnal cycle of precipitation over the Maritime Continent under modulation of MJO: Perspectives from cloud-permitting scale simulations. *J. Geophys. Res. Atmos.* **125**, (2020).
- Chappell, C. F. Quasi-stationary convective events. in *Mesoscale meteorology and forecasting* 289–310 (American Meteorological Society, 1986)
- Corfidi, S. F., Meritt, J. H., & Fritsch, J. M. Predicting the movement of mesoscale convective complexes. *Weather Forecast.* **11**, 41–46 (1996).
- VandenBerg, M. A., Coniglio, M. C., & Clark, A. J. Comparison of next-day convection-allowing forecasts of storm motion on 1-and 4-km grids. *Weather Forecast.* **29**, 878–893 (2014).

4. Line 225: How to determine the value of m when calculating the horizontal phase speed c based on Eq. (2)?

Thanks for your suggestions. The value of vertical wavenumber (m) or vertical wavelength (λ_z) is closely related to the vertical heating scale z_0 set in the analytical model (Li and Carbone 2015). z_0 was set as 1 km corresponding to typical boundary layer depths following the setting of the simple 2D linear land–sea-breeze model of Du and Rotunno (2015). The dominant vertical wavenumber and wavelength can be obtained by the cross

section of vertical motion from the analytical model, which are approximately 0.8×10^{-3} – $1.0 \times 10^{-3} \text{ m}^{-1}$ and 6–8 km respectively.

The statement was added in the revised manuscript (Lines 238–239, 432–442).

References:

Li, Y. & Carbone, R. E. Offshore propagation of coastal precipitation. *J. Atmos. Sci.* **72**, 4553–4568 (2015).

Du, Y. & Rotunno, R. Thermally driven diurnally periodic wind signals off the east coast of China. *J. Atmos. Sci.* **72**, 2806–2821 (2015).

5. Lines 350-354: The errors of the CMORPH need to be indicated herein.
Thanks for your suggestions. The errors of the CMORPH were added in the revised manuscript (Lines 383–386) as:
“Note that CMORPH has a negative bias at mid to high latitude land areas in the cool season due to unsatisfactory performance in detecting cold season precipitation and snowfall. Since OP is mainly distributed in tropical seas as well as during warm season, the above effects on the results is limited.”
6. Figure 2: There is no figure number for the subplots. No units can be found for the values with the triangles.
Done, thanks. The figure numbers and the units were added in the updated figure (new Fig. 2).
7. Figure 3: There are no units for latitude.
Done, thanks. The units were added in the updated figures (new Figs. 3 and 4).
8. Figure 4: In the figure caption, ‘D’ should be in the lower case.
Fixed. Thanks.
9. Figure 5: It is hard to read the words on the yellow arrows.
Thanks for your suggestions. To make the words more legible, we changed the yellow arrows to orange and dark yellow arrows, turned off the opacity of the arrows, and bolded the words on the arrows (Fig. R6 or new Fig. 5). The new Fig. 5 was updated in the revised manuscript.

a Diurnal rainfall offshore propagation at low latitudes

b Diurnal rainfall offshore propagation at middle latitudes

Fig. R6: Conceptual model of diurnal rainfall offshore propagation (OP). **a** OP at low latitudes in a high-moisture environment is characterized by strong rainfall and diurnal cycle, and is mainly caused by inertia-gravity waves forced by the terrain/land-sea thermal contrast. **b** OP at middle latitudes is characterized by weaker rainfall and diurnal cycle, and is significantly influenced by the direction of background winds under trapped land-sea breeze circulation conditions.

Response to Comments of Reviewer #2

Reviewer #2:

Overview: This study used satellite and reanalysis data to investigate diurnal offshore precipitation propagation on a global scale. Previous studies have typically focused on offshore propagation over one or a very few coasts therefore lack agreement from place to place. This study attempted to analyze the offshore precipitation characteristics depending on latitude, moisture and background wind from a globally unified perspective. The authors provided solid quantitative analysis to the inertia–gravity waves (IGW) and density current (DC), which adds up to the attempts of the community to reach a consensus in terms of physical mechanisms. I really appreciate the part where this study divided pronounced offshore propagation (POP) into pre-midnight and post-midnight samples, which helps explain the long-lasting discrepancy among previous studies where the offshore propagation speeds near shore are different based on different observations. However, there are a few issues that the authors may need to give additional analysis to make the arguments more concrete. The authors were also unclear about some methodology details. Overall, this study is interesting, well-written and of importance to our community. It could be considered to publish on Nature Communications after the following issues are addressed.

We appreciate the reviewer's comments and suggestions to significantly improve our manuscript. As suggested, we added additional analysis on the two-step POP1 in the revised manuscript, and further clarified some unclear methodologies. Our point-by-point response is given in blue as follows.

1. L261-264: The authors found that 29% of the total POP1 samples have two-step propagation. Since POP1 is defined as post-midnight samples, I would assume that its slower nearshore propagation starts after midnight. Then the questions are: Do the two-step POP1 samples have a nearshore propagation speed range similar to POP2 (i.e., 3-7 m/s)? If they do, is it related to DCs (because the authors mentioned in L254-255 that DCs generally propagate offshore before midnight)? If they don't, what causes the two-step POP1 events?

Thanks for your suggestions. We further examined the two-step POP1 samples as suggested. As shown in Hovmöller diagrams of POP1 (Fig. R7 or Supplementary Fig. 2), the one-step POP1 and two-step POP1 differ from each other. The one-step POP1 starts late at ~6:00 and maintains a high speed ($\sim 14 \text{ m s}^{-1}$, brown arrow) from coastline to far shore (Fig. R7a). In contrast, the two-step POP1 (Fig. R7b), however, starts ~3 h earlier than the one-step POP1 (~3:00) and is characterized by a slow speed near shore ($\sim 5 \text{ m s}^{-1}$, blue arrow) and a high speed far shore ($\sim 13 \text{ m s}^{-1}$, brown arrow). Such transition between near- and far-shore of the two-step POP1 is indeed similar to POP2 (Fig. 4c), as the reviewer guessed.

Fig. R7: Composite distance–time Hovmöller diagrams of the hourly precipitation deviation (shaded, mm) for one-step and two-step POP1. Two-step POP1 samples accounted for 29% of POP1 samples, 35 in total. One-step POP1 samples accounted for 71% of POP1 samples, 90 in total. The brown and blue arrows indicate the diurnal rainfall propagation driven by IGWs and density currents (DCs), respectively. The propagation speed transition for two-step POP1 is indicated by the grey dashed line. POP is pronounced offshore propagation with sufficient moisture ($q > 11 \text{ g kg}^{-1}$), and POP1 is post-midnight POP.

Specifically, the near-shore propagation speeds of two-step POP1 (blue circles) well match the DCs characteristic speed (sky blue range), and no trend in near-shore speed with latitude occurs (Fig. R8 or Supplementary Fig. 3). The results suggest that near-shore propagation in the two-step POP1 probably has a close association with DCs. In addition, the far-shore speed of the two-step POP1 (red circles) generally increases with latitude, with their least square fit line (red line) similar to that of the analytical solutions of inertia–gravity waves forced by the terrain/land–sea thermal contrast (hereafter referred to as IGWs, brown line), which is also similar to POP2.

Fig. R8: The differences of near- and far-shore propagation speed of the two-step POP1 samples. The far-shore propagation speed of the two-step POP1 samples (red circles) and analytical solutions of inertia-gravity wave forced by the terrain/land-sea thermal contrast (IGW, brown triangles), and near-shore propagation speed of the two-step POP1 samples (blue circles), IGW analytical solutions (grey triangles) and density current (DC) characteristic range (sky blue range, 3–7 m s⁻¹) with the latitude. The least square fit lines between the far-shore propagation speed and latitude for two-step POP1 ($r = 0.41$, $P < 0.02$) and IGWs ($r = 0.83$, $P < 0.005$) are indicated by red and brown lines, respectively. POP is pronounced offshore propagation with sufficient moisture ($q > 11 \text{ g kg}^{-1}$), and POP1 is post-midnight POP.

The relationship between the two-step POP1 and DCs is further explored through examining sensitivities of the advection effect of background winds (Fig. R9 or Supplementary Fig. 4). Most two-step POP1 samples (~71%) start before 4:00, while most one-step POP1 samples (~64%) start after 4:00. In one-step POP1 (brown dots), the proportions of offshore and onshore background winds are comparable (brown bar), indicating limited role of the background wind advection effect and dominant role of IGWs. In contrast, the offshore background winds are significantly more predominant in the two-step POPs, reaching 83% in POP2 (dark blue bar) and 69% in two-step POP1 (light blue bar). Such dependence on background wind direction is consistent with the advection properties of DCs. Therefore, two-step POP1 and POP2 may have similar mechanisms (DC) in near-shore areas.

Fig. R9: The relationship of background wind and onset phase between three types of POPs. Positive (negative) values of the averaged 850–300 hPa background wind indicate its offshore (onshore) component. The bars indicate the proportion of offshore and onshore wind for each type. POP is pronounced offshore propagation with sufficient moisture ($q > 11 \text{ g kg}^{-1}$), and POP1 (POP2) is post-midnight (pre-midnight) POP.

However, clear differences in starting phase for the two-step POP1 and POP2 are also found. The near-shore POP2 starts at evening ($\sim 21:00$, Fig. 4c) and is probably related to the cold pool outflows generated by afternoon land convection. However, the near-shore two-step POP1 starts late ($\sim 3:00$, Fig. R7b) and presumably has a weak relationship with the cold pools of afternoon convection. Previous studies suggest that most of the OP associated with DCs onset before midnight, whereas the rest of them start after midnight and are mainly related to land/mountain breeze fronts (e.g., Houze et al. 1981, Wapler and Lane 2012, Vincent and Lane 2016). In particular, Bai et al. (2020) documented the climatology of convective initiation (CI) in South China, and found post-midnight near-shore offshore propagation of CI. Since the offshore propagation of CI has already excluded the cool pool effect, the effect of land/mountain breezes might be important. Therefore, both two-step POP1 and POP2 in the near-shore areas are related to DCs, and specifically two-step POP2 is probably more related to convective cold pool outflows while two-step POP1 is probably more related to land/mountain breeze fronts.

The related Supplementary Figs. 2–4 and text were updated in the revised manuscript (Lines 265, 275, 282–285, 288–299, 321).

References:

Houze Jr, R. A., Geotis, S. G., Marks Jr, F. D. & West, A. K. Winter monsoon convection in the vicinity of north Borneo. Part I: Structure and time variation of the clouds and precipitation. *Mon. Weather Rev.* **109**, 1595–1614 (1981).

Wapler, K. & Lane, T. P. A case of offshore convective initiation by interacting land breezes near Darwin, Australia. *Meteorol. Atmos. Phys.* **115**, 123–137 (2012).

Vincent, C. L. & Lane, T. P. Evolution of the diurnal precipitation cycle with the passage of a Madden–Julian Oscillation event through the Maritime Continent. *Mon. Weather Rev.* **144**, 1983–2005 (2016).

Bai, L., Chen, G. & Huang, L. Convection initiation in monsoon coastal areas (South China). *Geophys. Res. Lett.* **47**, (2020).

2. Fig. 2 and L396: The authors calculated the least square fit between the hourly precipitation and the diurnal amplitude of the offshore propagation. One thing to note is that time series data may have temporal autocorrelation. Therefore, when calculating least square fit using time series data, the effective sample size may not equal to ‘n’ (Bretherton 1999), leading to different t and p values. This study used precipitation data, which typically have a very low temporal autocorrelation, so I would say that the results should not change much, especially when the p-values are so small. But a few words on this issue would be good just for prudence.

References:

Bretherton, C.S., Widmann, M., Dymnikov, V.P., Wallace, J.M. and Bladé, I., 1999. The effective number of spatial degrees of freedom of a time-varying field. *Journal of climate*, 12(7), 1990-2009.

Thanks for your suggestions. We used the method described by Bretherton et al. to estimate effective sample size, and recalculate the t and p values in Fig. 2 as suggested. The effective sample size (N) is estimated by:

$$N = n \frac{1-r_1r_2}{1+r_1r_2} \quad (1)$$

where r_1 and r_2 are the lag-one autocorrelations of each series, respectively.

The calculated effective sample sizes for OP, phase-locked pattern (PL) and SP are 95, 169 and 73 respectively, which are reduced to 21%, 51% and 75% of the original sample sizes. However, their confidence levels change little (still $P < 0.001$ for all three), as the reviewer guessed.

The related statement was added in the revised manuscript (the caption of Fig. 2 and Lines 451–455).

3. Some unclear technical/methodology details:
 - 1) L138-139: The authors only described how they identified offshore propagation with the hourly precipitation data in the method section. But how did they determine whether a month had offshore propagation or not using monthly sample?
Each “monthly sample” represents mean diurnal variation of rainfall averaged in the corresponding month of 1998–2015, which contains a diurnal cycle set of hourly precipitation data (e.g., all 3 LST data of long series were averaged into 3 LST data of a

sample). We determined whether the sample had offshore propagation or not using the method.

Considering that the expression “monthly” is indeed ambiguous and could easily be misinterpreted as no diurnal cycle information, we clarified it in the method section and explained how *a studied sample* was constructed instead of “monthly sample”.

The related statement was updated in the revised manuscript (Lines 136–137, 408–412).

2) L145-146: How did the author calculate the mean hourly precipitation for the four categories? Specifically, up to what distance offshore/onshore did the authors average the precipitation? Is the area averaged the same for the four categories?

Thanks for your suggestions. We calculated the mean hourly precipitation of each sample within the rectangle enclosed by the selected coastline and its offshore direction (perpendicular to the coastline) up to 250 km. The distance offshore we average the precipitation are the same for the four categories (250 km). But the averaged areas might be slightly different depending on the length of selected coastline. Nevertheless, the averaged areas for the four categories are highly similar (difference <5%, OP:104500 km², SP: 102500 km², PL: 104250 km² and weak or incoherent pattern (WI): 107500 km²), whose impact on the results is supposed to be limited.

The related statement was updated in the revised manuscript (Lines 150, 402–404).

3) Fig.3: Over how large the area did the authors define the background wind and specific humidity?

Thanks for your suggestions. The calculation area of the background wind and specific humidity of each sample was the same as that of the mean hourly precipitation above, within the rectangle enclosed by the selected coastline and its offshore direction up to 250 km, which was 105000 km² on average.

The related statement was updated in the revised manuscript (Lines 402–404).

Minor comments:

1. L139-142: It is unclear what the percentages in the parentheses are. Are they percentages of the time or coasts?

Thanks for your suggestions. They are percentages of the studied samples. We adjusted our expressions in the relevant texts in the revised manuscript (Lines 138–140) as:

“In general, OP occurs frequently in the tropics (<30°), where OP accounts for ~41% of all samples, especially in humid environments. In contrast, OP accounts for only ~26% of all samples at middle latitudes (30°–60°).”

2. Fig. 2: The orange dots with green contours are so overlapped with other dots. It would be good to make the dots of other colors more transparent (or use other better ways).

Thanks for your suggestions. To make the orange dots with green contours more visible, we made the dots of other colors 25% more transparent, the green contours 40% less transparent and 25% thicker, and all dots 25% smaller in Fig. R10a (new Fig. 2a). And, we also added green and orange lines in Fig. R10b, c (new Fig. 2b, c) to represent the proportion of the orange dots with green contours and that with brown contours, respectively.

The new Fig. 2 and related text at Lines 219, 317, 329 were updated in the revised manuscript.

Fig. R10: Precipitation and the diurnal amplitude per diurnal pattern. **a** Scatter diagram of the mean diurnal amplitude of rainfall versus mean hourly precipitation. OP, PL, and SP denote offshore propagation, phase-locked and shoreward propagation diurnal rainfall patterns, respectively. The orange dots with brown contours indicate OP samples with sufficient moisture ($q > 11 \text{ g kg}^{-1}$). The orange dots with green contours indicate OP samples at higher latitudes ($>30^\circ$). The weak or incoherent (WI) diurnal cycle is not shown. The mean hourly precipitation of each sample was calculated within the rectangle enclosed by the selected coastline and its offshore direction (perpendicular to the coastline) up to 250 km. The diurnal amplitude was derived from the amplitude of the first harmonic (diurnal cycle) in Fourier analysis. The black dashed line indicates that the mean

hourly precipitation equals the diurnal amplitude. The least square fit lines between the hourly precipitation and diurnal amplitude of the OP (orange line, $r = 0.84$, $P < 0.001$), PL (grey line, $r = 0.83$, $P < 0.001$), and SP diurnal patterns (blue line, $r = 0.89$, $P < 0.001$) are indicated with the slopes (k). During the computation of the confidence levels, the reduction of effective sample size due to the autocorrelations has been considered (see "Methods"). **b** Graph of the diurnal pattern proportion with mean hourly precipitation (mm). **c** Graph of the diurnal pattern proportion with mean diurnal amplitude of rainfall (mm). The triangles indicate mean hourly precipitation or mean diurnal amplitude for each diurnal pattern. The brown lines in **b**, **c** indicate OP samples with sufficient moisture ($q > 11 \text{ g kg}^{-1}$). The green lines in **b**, **c** indicate OP samples at higher latitudes ($>30^\circ$).

REVIEWERS' COMMENTS

Reviewer #1 (Remarks to the Author):

The authors have well addressed my previous comments and the overall quality of the revised manuscript has been improved significantly. The current version of the article can be accepted for publication.

Reviewer #2 (Remarks to the Author):

The authors have clearly responded to my comments, and have carefully edited the manuscript to make their arguments more concrete. I have no further comments and would like to recommend the acceptance of this manuscript to be published in Nature Communications.

Response to Comments of Reviewers

Reviewer #1:

The authors have well addressed my previous comments and the overall quality of the revised manuscript has been improved significantly. The current version of the article can be accepted for publication.

Response: We highly appreciate the reviewer for the valuable time for reviewing our manuscript and constructive comments to help significantly improve the quality of our work.

Reviewer #2:

The authors have clearly responded to my comments, and have carefully edited the manuscript to make their arguments more concrete. I have no further comments and would like to recommend the acceptance of this manuscript to be published in Nature Communications.

Response: We would like to sincerely thank the reviewer for the valuable time the reviewer has spent reviewing our manuscript and providing insightful comments which greatly improve the quality of our work.